# Global Phosphoproteomic Analysis Reveals the Defense and Response Mechanisms of Japonica Rice under Low Nitrogen Stress

**DOI:** 10.3390/ijms24097699

**Published:** 2023-04-22

**Authors:** Shupeng Xie, Hualong Liu, Tianze Ma, Shen Shen, Hongliang Zheng, Luomiao Yang, Lichao Liu, Zhonghua Wei, Wei Xin, Detang Zou, Jingguo Wang

**Affiliations:** 1Key Laboratory of Germplasm Enhancement, Physiology and Ecology of Food Crops in Cold Region, Ministry of Education, Northeast Agricultural University, Harbin 150030, Chinaxinweineau@163.com (W.X.); 2Suihua Branch of Heilongjiang Academy of Agricultural Science, Suihua 152052, China

**Keywords:** *Japonica* rice, phosphoproteomic, carbon metabolism, low nitrogen, photosynthetic nitrogen use efficiency

## Abstract

Nitrogen-based nutrients are the main factors affecting rice growth and development. As the nitrogen (N) application rate increased, the nitrogen use efficiency (NUE) of rice decreased. Therefore, it is important to understand the molecular mechanism of rice plant morphological, physiological, and yield formation under low N conditions to improve NUE. In this study, changes in the rice morphological, physiological, and yield-related traits under low N (13.33 ppm) and control N (40.00 ppm) conditions were performed. These results show that, compared with control N conditions, photosynthesis and growth were inhibited and the carbon (C)/N and photosynthetic nitrogen use efficiency (PNUE) were enhanced under low N conditions. To understand the post-translational modification mechanism underlying the rice response to low N conditions, comparative phosphoproteomic analysis was performed, and differentially modified proteins (DMPs) were further characterized. Compared with control N conditions, a total of 258 DMPs were identified under low N conditions. The modification of proteins involved in chloroplast development, chlorophyll synthesis, photosynthesis, carbon metabolism, phytohormones, and morphology-related proteins were differentially altered, which was an important reason for changes in rice morphological, physiological, and yield-related traits. Additionally, inconsistent changes in level of transcription and protein modification, indicates that the study of phosphoproteomics under low N conditions is also important for us to better understand the adaptation mechanism of rice to low N stress. These results provide insights into global changes in the response of rice to low N stress and may facilitate the development of rice cultivars with high NUE by regulating the phosphorylation level of carbon metabolism and rice morphology-related proteins.

## 1. Introduction

Rice (*Oryza sativa* L.) is one of the most important staple food crops in world; approximately 50% of the global population eats rice as a staple food [1,2]. Large amounts of nitrogen (N) are required for rice growth, and this often becomes a major limiting nutrient that affects rice productivity [3] To adapt to N starvation, plants have developed a variety of physiological and biochemical changes, including an increase in N uptake by high-affinity transporters, changing root architecture, suppressing growth and photosynthesis, and increasing the autophagy process [4,5,6]. Therefore, the research on the molecular mechanism of rice adapting to N starvation will help improve the nitrogen use efficiency (NUE) of rice and increase yield.

In recent years, many studies have been conducted on the adaptation mechanism of rice to nitrogen stress from multiple levels of transcription, translation, and metabolism [5,6,7,8,9,10]. Yang et al. [9] used RNA-Seq analysis and detected 28 TF families, with a total of 85 TFs under N deficiency, that play an important role in N deficiency response and plant growth. Previous studies have shown that changes in rice carbon (C) and N metabolism play an important role in how rice adapts to external N changes [5]. Proteomic analysis under low N conditions showed that the abundance of proteins involved in cell differentiation, cell wall modification, phenylpropanoid biosynthesis, and protein synthesis changed differently, which was an important reason for root morphological changes [6]. Some studies have obtained genes that are expected to significantly increase rice yield and NUE, including *DEP1*, *GRF4*, *NRT1.1B*, *NR2*, and *NAC42-NPF6.1*. Previous studies have found that the *OsLBD37/39-OsTCP19-OsDLT* pathway is the key to determining rice N response and tillering. It has also been found that the 29 bp Indel in the *OsTCP19* promoter region is a natural variation that determines the key regulatory role of different rice varieties on tiller number under low N levels, revealing the adaptation mechanism of rice to low N tolerance [7]. However, the protein modification mechanism of specific N response still needs to be further analyzed and constructed, and whether these important DMPs can participate in the regulation of low N tolerance needs to be further studied.

The post-translational modification of proteins is also an important field of protein chemistry research, playing an important role in plant growth and adaptation to biological and abiotic stresses [11,12,13]. Protein modification includes phosphorylation, glycosylation, methylation, etc [14]. Among them, phosphorylation is the most basic, common, and important mechanism for regulating and controlling protein activity and function. Protein phosphorylation is an important way to achieve post-translational modifications, which are involved in almost all plant life processes [15,16]. The phosphorylation process mainly refers to the process of transferring the γ-position phosphate group on adenosine triphosphate (ATP) to specific amino acid residues of the substrate protein under the catalysis of protein kinases. Studies have shown that protein phosphorylation occurs in the growth and development of plants and in the physiological and signal regulation processes in response to various stresses [17] Liu et al. [18] showed that the calcium-dependent signaling pathway that involves calcium-dependent protein kinase (CPK) can phosphorylate NLP7 to control its subcellular localization and, through the NO_3_^−^-CPK-NLP signaling pathway, affect plant root development and leaf morphology. Therefore, analyzing the changes in phosphorylation levels of key proteins can reveal important plant traits and the regulatory mechanism of metabolic pathways at the post-translational regulation level.

In this study, rice was exposed to low and control N conditions for 60 days. We determined morphological and physiological characteristics under low and control N conditions to clarify the effects of low nitrogen on morphological, physiological, and yield-related traits of rice. The integrated analysis of rice morphological and physiological characteristics and phosphoproteomic profiles allowed us to acquire more insight into the molecular mechanism of morphological and physiological changes along with yield formation in rice under low N conditions, which could be of great practical significance in increasing NUE.

## 2. Results

### 2.1. Low Nitrogen Stress Affects the Morphology, Physiology, and Growth Characteristics of Rice Leaves

Compared with control nitrogen, nitrogen deficiency had a negative effect on leaf biomass and leaf area. N concentration had no significant effect on intercellular CO2 concentration (Ci) compared to control N conditions, low N decreased Chlorophyll a (Chl a), Chlorophyll b (Chl b), photosynthetic rate (Pn), and stomatal conductance (gs). Compared with control nitrogen, N deficiency led to great decreases in the content of nitrogen and carbon. Low N increased C/N, NUE, and PNUE of the rice compared to control N. Compared with control N conditions, the low N treatments led to great decreases in grain yield, effective panicles, and plant height at the mature stage, while the low N treatments increased seed setting rate and 1000-grain weight (Table 1).

### 2.2. Phosphoprotein Identification and Phosphorylated Site Location

This study conducted a quantitative study of phosphorylated proteomics of rice leaves under low N conditions. In order to ensure a high degree of credibility of the results, we used the localization probability > 0.75 standard to filter the authentication data. A total of 6636 phosphorylation sites corresponding to 2925 phosphoproteins were identified, of which 5996 sites on 2731 proteins contained quantitative information (Figure 1a). The proportions of pS, pT, and pY sites were calculated as 91.2%, 8.6%, and 0.2%, respectively (Figure 1b).

### 2.3. Features of Phosphorylated Proteins in Response to Low-N Stress

The analysis of differentially modified proteins (DMPs) was performed under low and control N conditions, and the results are shown in Figure 2a. DMPs were selected by *p*-value and fold change (FC) with thresholds of FC > 1.5 and a *p*-value < 0.05. Compared to control N conditions, 258 proteins were differentially modified under low N conditions, with 123 up-regulated and 133 down-regulated. The results of subcellular structure location classification analysis are shown in Figure 2b. The subcellular structure of DMPs is divided into seven categories, among which ‘Nucleus’, ‘Chloroplast’, and ‘Cytoplasm’ were more numerous. All the identified significant changed phosphoproteins (258) were used for GO annotation. The distribution pie charts for biological process, cellular component, and molecular function are shown in Figure 2c. Within biological process, the enriched GO terms were ‘cellular processes, ‘response to stimulus’, and ‘metabolic processes’. Within cellular component, the enriched GO terms were ‘cell’, ‘organelle’, and ‘membrane’. Within molecular function, the enriched GO terms were ‘binding’, ‘catalytic activity’, and ‘transporter activity’.

### 2.4. KEGG Pathway and Protein Domain Analysis

The quantified proteins in this study were divided into four categories according to the quantification ratio: Q1 (0 < ratio < 0.667), Q2 (0.67 < ratio < 0.77), Q3 (1.3 < ratio < 1.5), and Q4 (ratio > 1.5). KEGG pathway analysis indicated that all quantified proteins were grouped into 12 functional classes (Figure 3a). Down-regulated DPMPs were significantly enriched in ‘Autophagy-other’, ‘Porphyrin and chlorophyll metabolism’, ‘Photosynthesis-antenna proteins’, and ‘Spliceosom’. Up-regulated DPMPs were significantly enriched in ‘Carbon fixation in photosynthetic organisms’, ‘ABC transporters’, ‘Pentose phosphate pathway’, ‘Glycolysis/Gluconeogenesis’, ‘Carbon metabolism’, ‘Galactose metabolism’, ‘Starch and sucrose metabolism’, and ‘beta-Alanine metabolism’. Consistent with the results of the previous transcription and metabolism analysis, carbon and nitrogen generation was significantly affected by low N stress [5]. Protein domain analysis shown that most of the up-regulated DPMPs have protein kinase domains (Figure 3b). This result indicates that protein kinases play an important role in the process of rice adaptation to low N stress.

### 2.5. Protein Kinases in Response to N Deficiency

Given that activation of protein kinases always depends on their auto-phosphorylation, protein kinases involved in a low N stress response could be identified by phosphoproteomic analysis. A total of 24 protein kinases were detected as differentially modified, and most protein kinases were up-regulated (Table 2). Levels of phosphorylation of two CPKs, *OsCPK10* was decreased and *OsCPK20*, were increased in response to low N stress. *OsRLCK109*, *OsRLCK118*, *OsRLCK213*, *OsRLCK278*, and *OsSERK2* were differently phosphorylated in response to low N stress, indicating that RLCKs might also play important roles in response to low N stress. Additionally, we found that some other MAPK subfamily proteins, including *OsMAPKKK*α, *OsMAPKKKε*, and *OsMAP3K.16*, might also be involved in response to low N stress. The stress-activated protein kinase 4 (*OsSAPK4*), a member of the sucrose nonfermenting1-related protein kinase2 (*SnRK2*) protein kinases, was dephosphorylated in response to low N stress. These results suggested that CPKs, RLCKs, and MAPK also contribute to the morphological and physiological responses to low N stress in rice.

### 2.6. An Overview of Response and Adaptive Mechanism of Rice under Low-N Stress

As shown in Table 1, under low N conditions, the balance of C and N metabolism was changed, the capacity of carbon metabolism relative to N metabolism was enhanced, and rice growth and development was limited. This study found that some proteins related to C metabolism and growth were phosphorylated under low N conditions (Figure 4). Phosphorylation of these proteins is closely related to C and N balance and agronomic traits under low N conditions. As shown in Figure 4, compared with control N conditions, 5 chloroplast development-proteins, 6 chlorophyll synthesis-proteins, 6 photosynthesis-proteins, 14 carbon metabolism-proteins, 8 phytohormone-related proteins and 7 morphology-related proteins were identified as DMPs under low N conditions. Additionally, the expression levels of these genes were determined using qRT-PCR analysis; of these, 14 genes, including FLN1, FLN2, and Cpn60β1, were down-regulated and LhcB and PPC4 were up-regulated under low N conditions (differential gene expression with credible ANOVA analysis and FC ≥ 1.5). This result indicates that the study of phosphoproteomics under low N conditions is important to better understand the adaptation mechanism of rice to low N stress.

## 3. Discussion

With the development of high-throughput sequencing technology and systems bioinformatics, many studies have analyzed the plant response and adaptation mechanism to low N by using different technologies, such as transcriptome, metabolome and proteomics [5,6,7,8,9,10,11]. At the same time, some epigenetics and post-translational modifications are widely used in the research of plant growth and development, yield, quality, and resistance, which greatly enriches the diversity of the research methods and promotes the research process in the plant field [19,20,21,22,23,24]. In this study, the phosphorylation of rice leaves under low N conditions was comprehensively analyzed. The gene expression analysis of some phosphorylated proteins showed that the phosphorylation modification of proteins was not consistent with the transcription level. Both of them were important biological activities that allowed rice to adapt to changes in the external environment. Compared to the control N conditions, 258 proteins were differentially modified under low N conditions, with 123 up-regulated and 133 down-regulated. These proteins are of great significance for understanding the protein modification mechanism of rice to adapt to low N stress and improve N uptake and utilization efficiency of rice.

### 3.1. Effects of Low Nitrogen Stress on Carbon Metabolism in Rice

C and N metabolism are two of the most important basic metabolic pathways in plants [25], and both are essential for rice growth and grain yield [25,26,27,28]. Chloroplast development, chlorophyll synthesis, and photosynthesis reflects C metabolism in plants, which is closely related to the formation of rice yield [9,29,30]. N is an integral part of the synthesis of green leaf, and chlorophyll is the main nitrogenous compound in leaves. In this study, chlorophyll content decreased significantly under low N conditions. Previous studies have shown that *OsFLN1* interacts with *OsTrxZ* to regulate chloroplast development in rice [31]. *FLN2* mutants also showed abnormal chloroplast development under high temperature conditions [32]. Huang et al. [33] showed that *OsCOMT* encodes caffeic acid 3-O-methyltransferase and is involved in the biosynthesis of melatonin. *OsCOMT* significantly delayed leaf senescence at the filling stage by inhibiting the degradation of chlorophyll and chloroplasts and improving photosynthetic efficiency. In this study, most chloroplast development and chlorophyll synthesis proteins were observed to be significantly down-regulated phosphorylated under low N conditions, which might have led to a decrease in chlorophyll content.

Photosynthesis reflects C metabolism in plants, which is closely related to the formation of rice yield [34,35,36]. Some studies have shown that, with the increase of N application level, the photosynthesis of plants is redundant [35,36]. The study of photosynthesis and carbon metabolism-related protein phosphorylation in rice under low niN conditions will help to further improve rice yield and NUE [29,30,37]. The Rubisco enzyme content is significantly positively correlated with N content in leaves, and the activity of Rubisco enzyme is closely related to the photosynthetic productivity of rice [38,39]. Li et al. [40] studies have shown that the main factor limiting the growth of Pn under high N is the reduction of Rubisco enzyme activity. The higher Rubisco activity under low N condition is the main reason for high PNUE. RbcL is a ribulose bisphosphate carboxylase large chain precursor, and in this study, RbcL (173S) was observed to be significantly down-regulated phosphorylated and RbcL (447T) was observed to be significantly up-regulated phosphorylated under low N conditions, which might have led to an increase in Rubisco enzyme activity [41]. *LFNR1* is an important chloroplast enzyme that functions as the final step of the photosynthetic electron transport chain. LIR1 and *LFNR1* form thylakoid protein complexes with *TIC62* and *TROL*. *LIR1* can increase the affinity of *LFNR1* and *TIC62*, regulate the adhesion of *LFNR1* to thylakoid membrane, and affect rice photosynthesis [42]. These results indicate that NUE and rice yield may be further improved by mediating phosphorylation of rice photosynthesis-related proteins.

### 3.2. Effects of Low Nitrogen Stress on Agronomic Traits of Rice

In this study, the plant height, effective panicles, and grain yield of rice were significantly decreased, and the seed setting rate and 1000-grain weight increased significantly under low nitrogen conditions, which was consistent with previous studies [43]. Phytohormones play an important role in synergistically regulating the growth and development of rice organs, nutrient absorption, C and N assimilation transport and distribution, and inducing defensive adaptation to stress [44,45,46]. In this study, we found that some phytohormones-related proteins were phosphorylated under low N conditions. Among them, *FLR1*, *SD8*, *DEP2*, *REM4.1*, *SERK1*, and *GF14c* are closely related to plant morphology. *FLRs* genes maintain plant type and pollen activity of rice, whereas *FLR1* mainly regulates pollen activity and plant height, and also affects male gametophyte development. Similar to FER, *FLRs* affect cell elongation and regulate plant height through the GA synthesis pathway [47]. Compared with the wild type, the plant height of the *sd8* mutant was significantly reduced, the flag leaf angle was smaller, the plant was more upright, and the yield per plant was not affected. The IAA content of sd8 was lower than that of wild type, and the application of IAA could restore the dwarf and erect leaf phenotype of *sd8* [48]. In addition, the grain development-related proteins *RLCK118* and *EIN2* were phosphorylated [49,50]. Zhou et al. [49] showed that compared with wild-type, *OsRLCK118*-silenced transgenic plants had increased leaf inclination angle, sensitivity to brassinolide treatment, and a decreased seed setting rate. In this condition, the grain length of *EIN2* transgenic plants increased, while the thousand-grain weight of *ein2* mutant decreased [50].

Notably, in this study, we also found that some proteins directly affecting plant and spike morphology were phosphorylated. The *CESA7* mutant is brittle in culm and leaf, easy to break when bent, and dwarfed. The plant height is 50% of the wild type, the leaves and stems are drooping, the number of tillers is reduced by 33%, the seed setting rate is reduced by 23%, the cellulose content in the stem is reduced to 40% of the wild type, and the hemicellulose content is 27% higher than the wild type [51]. Knockdown of *TAF2* in Kongyu 131 decreased the grain size and grain width of transgenic plants. Compared with the wild type, the number of longitudinal and transverse cells in the glume of the knockdown lines decreased by 10.9% and 10.2%, respectively, while the cell size only decreased slightly, and the length and width decreased by 2.5% and 2.2%, respectively, indicating that *TAF2* mainly controls rice grain size by affecting cell division. Knockdown of *TAF2* rescued the grain enlargement phenotype of pow1 but had little effect on the leaf angle enlargement phenotype [52]. *FLO2* plays a key role in regulating rice grain size and starch quality by affecting the accumulation of storage substances in endosperm; it may also be involved in heat tolerance during seed development [53]. These results indicate that rice regulates major agronomic traits and yield formation through phosphorylation of some plant morphological and phytohormonal-related proteins under low N conditions.

## 4. Materials and Methods

### 4.1. Plant Material and Growth Conditions

The experiment was conducted in 2021 at the Northeast Agricultural University, Heilongjiang Province, China during the rice growing season. After germination on moist filter paper, rice (*Oryza sativa* L.) seeds (cvar. Dongfu 114 *Japonica* China) were disinfected with 0.01% HgCl_2_, allowed to germinate at 28 °C for 2 d, and further cultured in a greenhouse (28/25 °C, 10 h day/14 h night). At the three-leaf heart stage, the seedlings were transferred to low nitrogen (13.33 ppm) and control nitrogen (40 ppm) using NH_4_NO_3_ as the N source; then, they were grown in hydroponics for 60 d under natural conditions. The hydroponics solution was formulated according to the method by Li et al. [40], which was amended by adding 1 mL of dicyanamide (nitrate inhibitor) per 1 L of nutrient solution. All treatments had fifteen replicates with a completely random design, and after 60 d (booting stage), the relevant indicators were determined.

### 4.2. Determination of Leaf Morphological and Physiological Characteristics

Leaves were harvested at the booting stage, with five biological replicates per treatment. Leaves’ length and width were measured with a ruler, and leaves’ area were calculated by length × width × 0.75. Then, the leaves were green removed at 105 °C for 30 min, dried at 80 °C to a constant weight, and their biomass were measured with a balance. Then, samples were powdered with a micro-pulverizer (FZ102, Tianjin, China), and an element analyzer (Vario MACRO cube, Hanau, Germany) was used to determined leaves’ C and N contents. NUE (g g^−1^) = Aboveground dry matter accumulation (g)/aboveground nitrogen accumulation (g). The Pn, Ci, gs and ETR of the top 2 leaves were determined by a LI-6400 portable photosynthetic instrument (PP Systems, USA). The concentrations Chla and Chlb were determined based on the Li et al. [40] method. PNUE (μmol g^−^s^−^) = Pn (μmol m^−2^s^−^)/the leaves N content (g m^−2^).

### 4.3. Determination of Yield and Plant Height at Maturity Stage

At maturity, grain yield and plant height were determined. The five representative plants per treatments were sampled to determine their yield and its components. The panicle number was recorded to determine the panicle number per plant. Filled and unfilled grains of the panicles were manually separated to measure the grain number per panicle and the seed setting rate. Randomly selected filled grains from each hill were used to determine the 1000-grain weight.

### 4.4. Protein Extraction and Digestion

Leaves were harvested at the booting stage from three biological replicates per treatment. The sample was chilled to −80 °C, and an appropriate amount of sample was weighed into a mortar precooled in liquid nitrogen. The mortars were used to fully grind the liquid nitrogen to powder. A total of 4 times the volume of powdered phenol extraction buffer (containing 10 mM dithiothreitol, 1% protease inhibitor, 1% phosphatase inhibitor) was added to each sample and sonicated. Equal volume of Tris balanced phenol was added and centrifuged at 4 °C and 5500× *g* for 10 min. The supernatant was taken, and 5 times the volume 0.1 M ammonium acetate or methanol was added and allowed to precipitate overnight. The precipitate was then washed with either methanol and acetone, respectively. Finally, the precipitate was redissolved with 8 M urea, and the protein concentration was determined by BCA kit (Thermo Fisher Scientific, MA, USA). For digestion, the protein solution was reduced using 5 mM dithiothreitol for 30 min at 56 °C and alkylated with 11 mM iodoacetamide for 15 min at room temperature in darkness. The protein sample was then diluted by adding 100 mM TEAB to urea concentration less than 2 M. Finally, trypsin was added at a 1:50 trypsin-to-protein mass ratio for the first digestion overnight and 1:100 trypsin-to-protein mass ratio for a second 4 h-digestion.

### 4.5. Mass Spectrometry Analysis

The tryptic peptides were dissolved in 0.1% formic acid (solvent A), then directly loaded onto a home-made reversed-phase analytical column (15-cm length, 75 μm i.d.). The gradient showed an increase from 6% to 23% solvent B (0.1% formic acid in 98% acetonitrile) over 26 min, 23% to 35% in 8 min, and climbing to 80% in 3 min, then holding at 80% for the last 3 min, all at a constant flow rate of 400 nL/min on an EASY-nLC 1000 UPLC system (Thermo Fisher Scientific, MA, USA).The peptides were subjected to NSI source followed by tandem mass spectrometry (MS/MS) in Q ExactiveTM Plus (Thermo Fisher Scientific, MA, USA) coupled online to the UPLC. The electrospray voltage applied was 2.0 kV. The m/z scan range was from 350 to 1800 for full scan, and intact peptides were detected in the Orbitrap at a resolution of 70,000. Peptides were then selected for MS/MS using NCE setting as 28, and the fragments were detected in the Orbitrap at a resolution of 17,500. A data-dependent procedure that alternated between one MS scan and 20 MS/MS scans with 15.0 s dynamic exclusion. Automatic gain control (AGC) was set at 5 × 10^−4^. Fixed first mass was set as 100 m/z.

### 4.6. Database Search

The resulting MS/MS data were processed using the Maxquant search engine (Matrix Science, London, UK, v.1.5.2.8). Tandem mass spectra were searched against rice uniprot database concatenated with reverse decoy database (*Oryza_sativa*_subsp._*japonica*_ 39947). Trypsin/*P* was specified as a cleavage enzyme, allowing up to 4 missing cleavages. The mass tolerance for precursor ions was set as 20 ppm in the first search and 5 ppm in the main search, and the mass tolerance for fragment ions was set as 0.02 Da. Carbamidomethyl on Cys was specified as fixed modification and acetylation modification, and oxidation on Met were specified as variable modifications. FDR was adjusted to <1% and minimum score for modified peptides was set >40.

### 4.7. Bioinformatics Analysis

The differential expression modification proteins (DMPs) analysis across samples was conducted in the edgeR package (http://www.r-project.org/, 3 April 2021) to obtain DMPs (*p*-value < 0.05, FC > 1.5). Subcellular localization annotation of the submitted proteins was performed using the software wolfpsort, which predicts subcellular localization. For prokaryotes, we used CELLO software to predict the subcellular structure of their proteins. Gene ontology (GO) and Kyoto Encyclopedia of Genes and Genomes (KEGG) analysis of the DEGs were implemented to identify significantly enriched GO terms and metabolic pathways. GO enrichment analysis was conducted in the GOSeq R program (http://www.r-project.org/, 3 April 2021), and KEGG pathway analysis was conducted using the KEGG Orthology program (http://kobas.cbi.pku.edu.cn/, 3 April 2021) [54].

### 4.8. qRT-PCR Analysis

Three samples of rice first leaf were examined after low-N and control-N treatment for 60; three biological replicates per sample were used for qRT-PCR analyses. The first strand cDNA was synthesized using the Prime Script RT Master Mix (Takara). Real-time PCR was performed with SYBR Premix Ex Taq II (Takara) according to the manufacturer’s protocol in Applied Biosystems QuantStudio 3 (Thermo Fisher Scientific, Germany); transcript specific primers are listed in Appendix A. Relative quantification analysis was performed with a relative standard curve for threshold values (CT). qRT-PCR data was standardized using *OsACTIN1* as an internal reference.

## 5. Conclusions

In this study, we compared morphological, physiological, and yield-related traits and phosphoproteomic profiles under low and control N conditions. Compared with control N conditions, a total of 258 DMPs were identified under low N conditions. Research results suggest that N may affect rice growth and yield formation by regulating biological processes such as chloroplast development, chlorophyll synthesis, photosynthesis, and metabolism of C and phytohormones. This study elucidated the protein modification mechanism underlying the low N-mediated plant morphology and physiological changes and yield formation and provides new insights into improving nitrogen use efficiency in rice. In future studies, it will be important to determine the role of these DMPs in low N for improving rice yield and NUE.

## Figures and Tables

**Figure 1 ijms-24-07699-f001:**
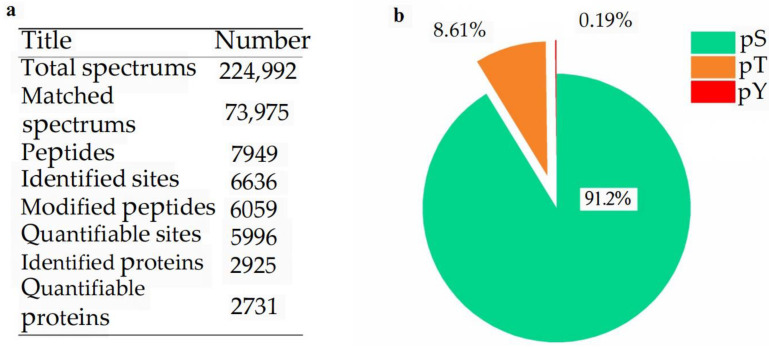
Overview of phosphoprotein identification. (**a**), phosphoprotein identification; (**b**), phosphorylated site type analysis.

**Figure 2 ijms-24-07699-f002:**
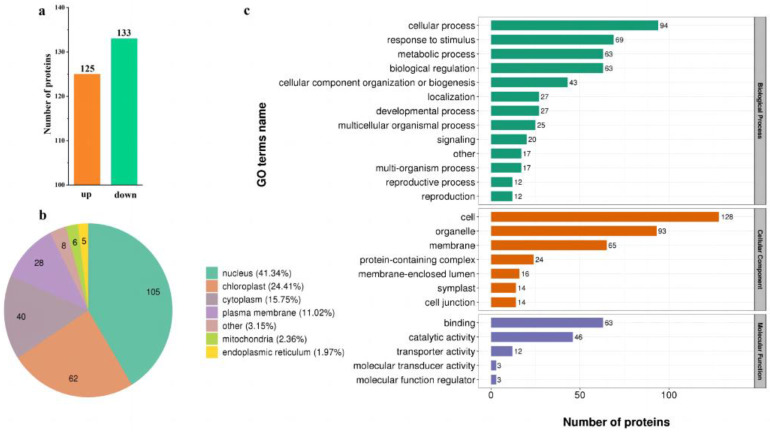
Phosphorylation level and phosphoproteomic analysis. (**a**), The number of phosphorylated proteins; (**b**), subcellular localization analysis of phosphorylated proteins; (**c**), functional classification of the significantly changed phosphoproteins under different treatment by Go analysis.

**Figure 3 ijms-24-07699-f003:**
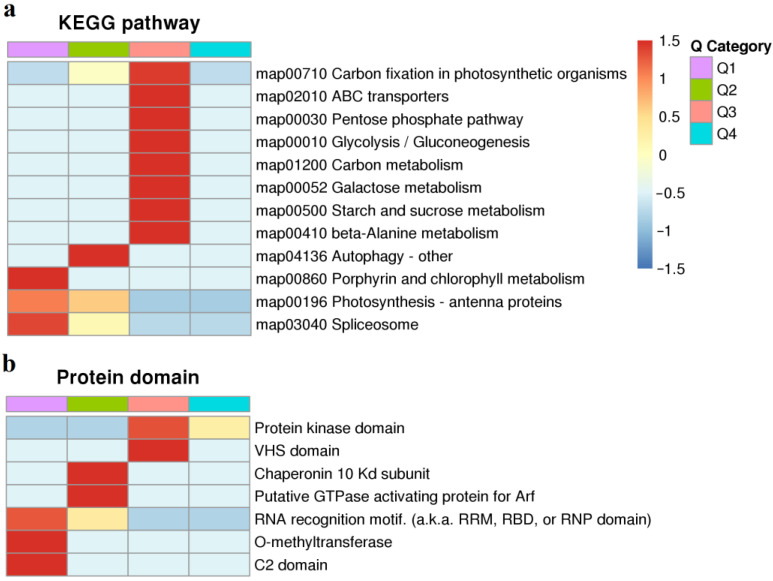
KEGG pathway and protein domain analysis. (**a**), KEGG pathway analysis; (**b**), protein domain analysis.

**Figure 4 ijms-24-07699-f004:**
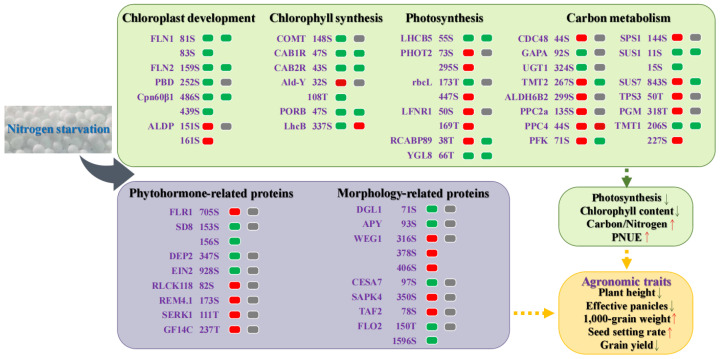
A summary of differences in morphysiology and phosphoproteomics in response to nitrogen starvation in rice. The red squares indicate up-regulation, the green squares indicate down-regulation, and the gray squares indicate no significant difference. The first column is protein phosphorylation data and the second column is qRT-PCR data. qRT-PCR data were standardized using *OsACTIN1* as an internal reference, and data are presented as means (n = 3). The up arrows indicate increases and the down arrows indicate decreases under low nitrogen conditions.

**Table 1 ijms-24-07699-t001:** Morphology, physiology, and growth response to low nitrogen conditions. Values labeled with different letters in same row indicate significant difference between the nitrogen treatments; data are presented as means ± S.D. (*n* = 5). *p* values of the ANOVAs are indicated.

Treatments	Low Nitrogen	Control Nitrogen
Leaf biomass (g)	1.58 ± 0.20 b	2.35 ± 0.17 a
Leaf area (cm^2^)	276.53 ± 18.38 b	407.18 ± 11.65 a
Chlorophyll a (Chl a, mg g^−1^)	0.93 ± 0.05 b	1.76 ± 0.12 a
Chlorophyll b (Chl b, mg g^−1^)	0.42 ± 0.02 b	0.82 ± 0.07 a
Intercellular CO2 concentration (Ci, μmol mol^−1^)	268.55 ± 8.13 a	274.56 ± 16.95 a
Photosynthetic rate (Pn, μmol m^−2^ s^−1^)	19.27 ± 0.72 b	22.51 ± 1.21 a
Stomatal conductance (gs, mol m^−2^ s^−1^)	639.23 ± 12.37 b	723.56 ± 24.13 a
Electron transport rate (ETR, μmol m^−2^ s^−1^)	11.83 ± 0.19 b	12.29 ± 0.18 a
N content	3.53 ± 0.21 c	4.72 ± 0.32 b
C content	36.11 ± 0.21 b	42.16 ± 0.43 a
Carbon/Nitrogen (C/N)	10.23 ± 0.14 a	8.93 ± 0.12 b
Nitrogen use efficiency (NUE, g g^−1^)	48.98 ± 3.14 a	31.21 ± 3.25 b
Photosynthetic nitrogen use efficiency (PUNE, μmol g^−1^ s^−1^)	9.55 ± 0.58 a	8.26 ± 0.18 b
Grain yield (g plant^−1^)	19.34 ± 1.78 b	27.59 ± 0.91 a
Effective panicles	8.18 ± 0.35 b	12.33 ± 0.35 a
Grain per panicle	121.32 ± 2.54 a	119.92 ± 1.29 a
Seed setting rate (%)	93.33 ± 1.78 a	85.51 ± 2.78 b
1000-grain weight (g)	24.12 ± 0.43 a	22.65 ± 0.43 b
Plant height at the mature stage (cm)	86.16 ± 2.18 b	97.34 ± 3.34 a

**Table 2 ijms-24-07699-t002:** Differentially phosphorylated kinases in response to N starvation in rice.

ID	Modified Sequence	LN/HN Ratio	Gene Name
LOC_Os03g57450	VS(0.001)S(0.999)AGLLVGSVLK	0.56	*OsCPK10*
LOC_Os07g38120	DGS(1)LQLTTTQ	1.513	*OsCPK20*
	FT(0.002)S(0.993)LS(0.005)LK	1.536	
	FTS(0.001)LS(0.999)LK	1.54	
LOC_Os04g38480	LMDYKDT(0.999)HVT(0.86)T(0.141)AVR	2.368	*OsSERK2*
LOC_Os03g24930	LS(0.004)S(0.996)MTNSPASSVAGAAEGGK	2.375	*OsRLCK109*
LOC_Os03g60710	NFRPDS(1)VLGEGGFGSVYK	2.634	*OsRLCK118*
LOC_Os06g46330	AT(0.023)S(0.891)S(0.079)S(0.006)S(0.001)LLTSIMAR	1.566	*OsRLCK213*
LOC_Os09g36320	SIS(1)SLYEER	1.668	*OsRLCK278*
LOC_Os11g10100	LSETS(0.001)VS(0.999)PR	0.58	*OsMAPKKKα*
LOC_Os04g56530	LDHHHS(0.917)S(0.083)GSLQSLQADADR	0.575	*OsMAPKKKε*
LOC_Os04g35700	VQS(1)PY(0.001)GS(0.999)PK	0.282	*OsMAP3K.16*
LOC_Os06g05520	FLTAS(0.001)GT(0.999)FKDGELR	1.967	*OsMKK1*
LOC_Os01g64970	EVHAS(1)GELR	1.563	*OsSAPK4*
LOC_Os01g42294	TTTEESEEGVRGT(0.003)S(0.997)EEER	1.65	*OsRPK1*
LOC_Os01g28730	S(0.933)FT(0.067)HINEDAALESPKEE	1.696	*OsRKF3*
LOC_Os11g11490	S(0.116)GT(0.884)DQFDLTDTD	1.685	*OsCRR4*
LOC_Os05g47560	TINES(1)MDELSSQSK	1.674	*OsSTN7*
	TINESMDELS(0.028)S(0.965)QS(0.007)K	1.582	
	VVRT(1)INES(1)MDELSSQSK	2.062	
LOC_Os09g23570	NADVDDFDS(0.002)VS(0.998)Q	1.978	
LOC_Os06g43840	VAS(1)RENISPK	0.53	
LOC_Os07g43560	HSTAMS(1)LNDVTVTEPEPR	2.051	
	LSLSYS(0.867)S(0.133)R	1.775	
	SDS(0.979)S(0.021)SLDEILR	1.812	
LOC_Os03g03570	KPVES(1)PGVATAVVLR	1.675	
LOC_Os07g43570	NRS(0.999)YT(0.001)ETMDVPLPSGPHSSITELEPR	1.514	
	RLS(0.998)NCS(0.002)NQGLGQLK	1.548	
LOC_Os03g27990	NEPLTLRPIAS(1)GK	0.533	
LOC_Os12g01200	ELPSSIHHLMS(1)K	1.796	
	LGS(1)FFSEVATESAHR	2.033	

## Data Availability

Not applicable.

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
