# Peer review of "Global Phosphoproteomic Analysis Reveals the Defense and Response Mechanisms of Japonica Rice under Low Nitrogen Stress"

_ijms, 2023, doi:10.3390/ijms24097699_

Round 1

Reviewer 1 Report

Dear authors,

thanks for submitting your interesting paper on obviously hard scientific work.

In general - it needs improvements before I can go into details.

- Generally, the text is not easy to read and understand - and I am not used to some terms (eg morphysiological).
- In general - the paper really needs to be re-arranged and worked over.

- finally there should be a clear "red string" from introduction to results and conclusion.

- Please re-arrange the order of paragraphs. Usually Materials&Methods follow the introduction.
Clarify precisely the aim of your research, why you made decisions on methods and your findings on results.
Materials and Methods MUST be clear in all details - e.g. the amount of fertilization, the structure of your measurements and devices...

- and this structure should be kept in results. Probably add discussion to results - you present a wide range of results (to help readers follow your thoughts and arguments)
- Conclusions is simply really poor

- Scientific names are in italic (not the L. for Linnaeus). And you should provide the botanic family once (in the introduction).
Maybe some explanation might be helpful why your research is of relevance for agronomic practices (is it for wet rice production)

- Reflect on maybe submitting two papers - one with the focus on eco-physiological traits and one on molecular physiology

- Please do not use abbreviations before defining them

- Maybe I missed the information - did you involve a control in comparison to low and high Nitrogen supply?

Please take time and find critical support - your work is worth the efforts

All the best - and hopefully I can see a revised paper soon!

Author Response

Thanks for editor and reviewers comments concerning our manuscript entitled "Global Phosphoproteomic Analysis Reveals the Defense and Response Mechanisms of Japonica Rice under Low Nitrogen Stress". These comments are valuable and helpful for us to revise our manuscript and guide our future research. We had carefully revised the paper according to these comments, and we hope this revision would be approved by you. The main changes in the manuscript and responses to these comments are listed below.

Reviewer1 Comments:

In general - it needs improvements before I can go into details.

Response: Thanks to reviewer for the comments. We have revised according to the reviewer’s suggestion.

- Generally, the text is not easy to read and understand - and I am not used to some terms (eg morphysiological).

Response: Thanks to reviewer for the comments.We are very sorry for this imprecise description, we have revised some terms according to the reviewer’s suggestion.

- In general - the paper really needs to be re-arranged and worked over.

Response: Thanks to reviewer for the comments.We have re-arranged and worked in full text according to the reviewer’s suggestion.

- finally there should be a clear "red string" from introduction to results and conclusion.

Response: Thanks to reviewer for the comments.We have revised according to the reviewer’s suggestion. 

- Please re-arrange the order of paragraphs. Usually Materials&Methods follow the introduction.

Response: Thanks to reviewer for the comments. Our manuscripts refer to the requirements of the IJMS journal submission format, and the Materials and Methods follow the discussion. Thank you again for your valuable comments.

Clarify precisely the aim of your research, why you made decisions on methods and your findings on results.

Response: Thanks to reviewer for the comments. We are very sorry about the unclear purpose of the study. We have revised in the abstract, introduction and discussion section according to the reviewer’s suggestion.

Materials and Methods MUST be clear in all details - e.g. the amount of fertilization, the structure of your measurements and devices...

Response: Thanks to reviewer for the comments. We have revised according to the reviewer’s suggestion.

“At the three-leaf heart stage, the seedlings were transferred to low nitrogen (13.33 ppm) and control nitrogen (40 ppm) using NH4NO3 as the N source, and grown in hydropoics for 60d under natural conditions.”

And, we have also revised measurement parameters and instrument information according to the reviewer’s suggestion.

- and this structure should be kept in results. Probably add discussion to results - you present a wide range of results (to help readers follow your thoughts and arguments)

Response: Thanks to reviewer for the comments. According to the requirements of IJMS journals, the results and discussions need to be separated.

- Conclusions is simply really poor

Response: Thanks to reviewer for the comments.We have Reorganized the conclusion part according to the reviewer’s suggestion.

- Scientific names are in italic (not the L. for Linnaeus). And you should provide the botanic family once (in the introduction).

Response: Thanks to reviewer for the comments.We have revised according to the reviewer’s suggestion.

Maybe some explanation might be helpful why your research is of relevance for agronomic practices (is it for wet rice production)

Response: Thanks to reviewer for the comments.We have revised according to the reviewer’s suggestion.

- Reflect on maybe submitting two papers - one with the focus on eco-physiological traits and one on molecular physiology

Response: Thanks to reviewer for the comments. In the next study, we will deeply consider your suggestions.

- Please do not use abbreviations before defining them

Response: Thanks to reviewer for the comments. We have revised in full text according to the reviewer’s suggestion.

- Maybe I missed the information - did you involve a control in comparison to low and high Nitrogen supply?

Response: Thanks to reviewer for the comments. We are very sorry for the confusing description of the N supply, there are two N supplies in this study, the low N conditions and control N conditions. And we have revised according to the reviewer’s suggestion.

Please take time and find critical support - your work is worth the efforts

All the best - and hopefully I can see a revised paper soon!

Response: Thanks to reviewer for the comments.

Reviewer 2 Report

the paper entitled “Global Phosphoproteomic Analysis Reveals the Defense and Response Mechanisms of Japonica Rice under Low Nitrogen Stress” provided novelties regarding the mechanisms of defense that occurred within Japonica Rice Under abiotic stress.  The paper is slightly well written with a decent technical level. Some important and major queries should be addressed. In addition to these general comments, below are the specific comments about the changes necessary to the text.

Title

Why did you specifically choose the low Nitrogen Stress? It would be also interesting to include the Nitrogen Excess (Very high rates of N) kind of stress and do a holistic comparative study.

Abstract

The main key messages from this study should be highlighted In this section.  

Introduction

The section is considerably short. It is important to mention all recent findings related to defense and response mechanisms.

The objectives of the study should be detailly covered in this section.

Discussion

This section should be further improved with relevant findings.

Conclusion

I would recommend including study perspectives for possible outcomes.

Author Response

Thanks for editor and reviewers comments concerning our manuscript entitled "Global Phosphoproteomic Analysis Reveals the Defense and Response Mechanisms of Japonica Rice under Low Nitrogen Stress". These comments are valuable and helpful for us to revise our manuscript and guide our future research. We had carefully revised the paper according to these comments, and we hope this revision would be approved by you. The main changes in the manuscript and responses to these comments are listed below.

Reviewer 2 Comments:

the paper entitled “Global Phosphoproteomic Analysis Reveals the Defense and Response Mechanisms of Japonica Rice under Low Nitrogen Stress” provided novelties regarding the mechanisms of defense that occurred within Japonica Rice Under abiotic stress.  The paper is slightly well written with a decent technical level. Some important and major queries should be addressed. In addition to these general comments, below are the specific comments about the changes necessary to the text.

Title

Why did you specifically choose the low Nitrogen Stress? It would be also interesting to include the Nitrogen Excess (Very high rates of N) kind of stress and do a holistic comparative study.

Response: Thanks to reviewer for the comments. From the perspective of weight loss and environmental protection, we only studied the adaptation mechanism of rice to low nitrogen.

Abstract

The main key messages from this study should be highlighted In this section.  

Response: Thanks to reviewer for the comments.We have revised according to the reviewer’s suggestion.

Introduction

The section is considerably short. It is important to mention all recent findings related to defense and response mechanisms.

The objectives of the study should be detailly covered in this section.

Response: Thanks to reviewer for the comments.We have revised according to the reviewer’s suggestion.

Discussion

This section should be further improved with relevant findings.

Response: Thanks to reviewer for the comments.We have revised according to the reviewer’s suggestion.

Conclusion

I would recommend including study perspectives for possible outcomes.

Response: Thanks to reviewer for the comments.We have revised according to the reviewer’s suggestion.

Reviewer 3 Report

Review of ijms-2324162

 Global Phosphoproteomic Analysis Reveals the Defense and Response Mechanisms of Japonica Rice under Low Nitrogen Stress

 Shupeng Xie, Hualong Liu, Tianze Ma, Shen Shen, Hongliang Zheng, Luomiao Yang, Lichao Liu, Zhong-hua Wei, Wei Xin, Detang Zou and Jingguo Wang

The authors wished to study the effect of Nitrogen deficiency on rice phosphoproteomes, since the activities of many proteins are regulated post-transcriptionally by phosphorylation. They therefore subjected rice cv. 'Dongfu 114' Japonica China to reduced N and compared their physiology, transcriptomes and phosphoproteomes to those of plants grown under control conditions. They detected 258 proteins that showed differential phosphorylation under low N.  They then determined what processes they were involved in, and concluded that differential phosphorylation plays an important role in acclimation to low N.

Overall, it seems that the data is solid and worth sharing with the plant community after correcting several problems, as noted below.

The caption to table 1 should indicate whether the ± values are standard deviation or standard error. It should also indicate the statistical test that was used.

Please explain acronymns such as NUE and PNUE the first time they are mentioned in the text.

Section 4.8 on qRT-PCR analysis should describe how the RNA was extracted and quantified. It should also indicate how many biological and technical replicates were performed.

The caption for figure 4 should show the mean ± standard deviations for the qRT-PCR data, and the caption should indicate how many replicates were performed and that OsACTIN1 was used as reference gene.

Because the lines aren’t numbered I can’t address specific problems. However, there are numerous instances of using the wrong tense such as past versus present (in this report you are reporting what you did, not what is currently happening) and singular vs plural. Also many sentences such as “The modification of proteins involved in chloroplast development-proteins, chlorophyll synthesis-proteins, photosynthesis-proteins, carbon metabolism-proteins, phytohormone-related proteins and morphology-related proteins were differentially altered, which was an important reason for changes in rice agronomic traits.” along with many others must be rewritten for clarity and grammar.

“differential modification” should be changed to “differentially modified”

Author Response

Thanks for editor and reviewers comments concerning our manuscript entitled "Global Phosphoproteomic Analysis Reveals the Defense and Response Mechanisms of Japonica Rice under Low Nitrogen Stress". These comments are valuable and helpful for us to revise our manuscript and guide our future research. We had carefully revised the paper according to these comments, and we hope this revision would be approved by you. The main changes in the manuscript and responses to these comments are listed below.

Reviewer 3 Comments:

The authors wished to study the effect of Nitrogen deficiency on rice phosphoproteomes, since the activities of many proteins are regulated post-transcriptionally by phosphorylation. They therefore subjected rice cv. 'Dongfu 114' Japonica China to reduced N and compared their physiology, transcriptomes and phosphoproteomes to those of plants grown under control conditions. They detected 258 proteins that showed differential phosphorylation under low N.  They then determined what processes they were involved in, and concluded that differential phosphorylation plays an important role in acclimation to low N.

Overall, it seems that the data is solid and worth sharing with the plant community after correcting several problems, as noted below.

Response: Thanks to reviewer for the comments.

The caption to table 1 should indicate whether the ± values are standard deviation or standard error. It should also indicate the statistical test that was used.

Response: Thanks to reviewer for the comments. We have revised according to the reviewer’s suggestion. (Table 1)

Please explain acronymns such as NUE and PNUE the first time they are mentioned in the text.

Response: Thanks to reviewer for the comments.We have revised according to the reviewer’s suggestion.

Section 4.8 on qRT-PCR analysis should describe how the RNA was extracted and quantified. It should also indicate how many biological and technical replicates were performed.

Response: Thanks to reviewer for the comments. We have revised according to the reviewer’s suggestion. (line 381-382)

The caption for figure 4 should show the mean ± standard deviations for the qRT-PCR data, and the caption should indicate how many replicates were performed and that OsACTIN1 was used as reference gene.

Response: Thanks to reviewer for the comments.We have revised according to the reviewer’s suggestion. (Figure 4)

Because the lines aren’t numbered I can’t address specific problems. However, there are numerous instances of using the wrong tense such as past versus present (in this report you are reporting what you did, not what is currently happening) and singular vs plural. Also many sentences such as “The modification of proteins involved in chloroplast development-proteins, chlorophyll synthesis-proteins, photosynthesis-proteins, carbon metabolism-proteins, phytohormone-related proteins and morphology-related proteins were differentially altered, which was an important reason for changes in rice agronomic traits.” along with many others must be rewritten for clarity and grammar.

Response: Thanks to reviewer for the comments. We are very sorry about the line number and grammar error. We have revised according to the reviewer’s suggestion.

“differential modification” should be changed to “differentially modified”

Response: Thanks to reviewer for the comments.We have revised according to the reviewer’s suggestion.

Round 2

Reviewer 1 Report

Thanks and respect to the authors - the paper has been improved significantly. Thanks for your responses.

I suggest to do some minor corrections:

Please check the term physiologica - it should either be physiology or physiological (several times in the text)

line 46 - L. is NOT italic (as I already commented in my first feedback)

line 119 - correct identifification (identification)

Figures - they seem to be inserted from somewhere else. Does this require a quotation or referenc? Or your own source?

line 272 correct to phytohormones

line 308: remove cultivars; according to the International Code cultivars are in single quotation marks OR cvar. (followed by the name) - I suggest to simply remove the quotation marks.

Line 318 - I suggest to write Leaf instead of Leaves

Line 335 - remove the of (does not fit the new text)

Line 375 Oryza sativa ... japonica in italic.

Author Response

Thanks for editor and reviewers comments concerning our manuscript entitled "Global Phosphoproteomic Analysis Reveals the Defense and Response Mechanisms of Japonica Rice under Low Nitrogen Stress". These comments are valuable and helpful for us to revise our manuscript and guide our future research. We had carefully revised the paper according to these comments, and we hope this revision would be approved by you. The main changes in the manuscript and responses to these comments are listed below.

Reviewer1 Comments:

Thanks and respect to the authors - the paper has been improved significantly. Thanks for your responses.

I suggest to do some minor corrections:

Please check the term physiologica - it should either be physiology or physiological (several times in the text)

Response: Thanks to reviewer for the comments. We have revised according to the reviewer’s suggestion.

line 46 - L. is NOT italic (as I already commented in my first feedback)

Response: Thanks to reviewer for the comments. We have revised according to the reviewer’s suggestion.

line 119 - correct identifification (identification)

Response: Thanks to reviewer for the comments. We have revised according to the reviewer’s suggestion.

Figures - they seem to be inserted from somewhere else. Does this require a quotation or referenc? Or your own source?

line 272 correct to phytohormones

Response: Thanks to reviewer for the comments. We have revised according to the reviewer’s suggestion.

line 308: remove cultivars; according to the International Code cultivars are in single quotation marks OR cvar. (followed by the name) - I suggest to simply remove the quotation marks.

Response: Thanks to reviewer for the comments. We have revised according to the reviewer’s suggestion.

Line 318 - I suggest to write Leaf instead of Leaves

Response: Thanks to reviewer for the comments. We have revised according to the reviewer’s suggestion.

Line 335 - remove the of (does not fit the new text)

Response: Thanks to reviewer for the comments. We have revised according to the reviewer’s suggestion.

Line 375 Oryza sativa ... japonica in italic.

Response: Thanks to reviewer for the comments. We have revised according to the reviewer’s suggestion.

Reviewer 2 Report

The authors did substantial improvement to their manuscript and i believe it is mature enough to be accepted for publication.

Author Response

Thanks for editor and reviewers comments concerning our manuscript entitled "Global Phosphoproteomic Analysis Reveals the Defense and Response Mechanisms of Japonica Rice under Low Nitrogen Stress". These comments are valuable and helpful for us to revise our manuscript and guide our future research.